# Convergence rates of a partition based Bayesian multivariate density estimation method

**Linxi Liu** *
Department of Statistics
Columbia University
ll3098@columbia.edu

**Dangna Li**
ICME
Stanford University
dangna@stanford.edu

**Wing Hung Wong**
Department of Statistics
Stanford University
whwong@stanford.edu

## Abstract

We study a class of non-parametric density estimators under Bayesian settings. The estimators are obtained by adaptively partitioning the sample space. Under a suitable prior, we analyze the concentration rate of the posterior distribution, and demonstrate that the rate does not directly depend on the dimension of the problem in several special cases. Another advantage of this class of Bayesian density estimators is that it can adapt to the unknown smoothness of the true density function, thus achieving the optimal convergence rate without artificial conditions on the density. We also validate the theoretical results on a variety of simulated data sets.

## 1   Introduction

In this paper, we study the asymptotic behavior of posterior distributions of a class of Bayesian density estimators based on adaptive partitioning. Density estimation is a building block for many other statistical methods, such as classification, nonparametric testing, clustering, and data compression.

With univariate (or bivariate) data, the most basic non-parametric method for density estimation is the histogram method. In this method, the sample space is partitioned into regular intervals (or rectangles), and the density is estimated by the relative frequency of data points falling into each interval (rectangle). However, this method is of limited utility in higher dimensional spaces because the number of cells in a regular partition of a $p$-dimensional space will grow exponentially with $p$, which makes the relative frequency highly variable unless the sample size is extremely large. In this situation the histogram may be improved by adapting the partition to the data so that larger rectangles are used in the parts of the sample space where data is sparse. Motivated by this consideration, researchers have recently developed several multivariate density estimation methods based on adaptive partitioning [13, 12]. For example, by generalizing the classical Pólya Tree construction [7, 22] developed the Optional Pólya Tree (OPT) prior on the space of simple functions. Computational issues related to OPT density estimates were discussed in [13], where efficient algorithms were developed to compute the OPT estimate. The method performs quite well when the dimension is moderately large (from 10 to 50).

The purpose of the current paper is to address the following questions on such Bayesian density estimates based on partition-learning. Question 1: what is the class of density functions that can be "well estimated" by the partition-learning based methods. Question 2: what is the rate at which the posterior distribution is concentrated around the true density as the sample size increases. Our main contributions lie in the following aspects:

- We impose a suitable prior on the space of density functions defined on binary partitions, and calculate the posterior concentration rate under the Hellinger distance with mild assumptions. The rate is adaptive to the unknown smoothness of the true density.

- For two dimensional density functions of bounded variation, the posterior contraction rate of our method is $n^{-1/4}(\log n)^3$.

- For Hölder continuous (one-dimensional case) or mixture Hölder continuous (multi-dimensional case) density functions with regularity parameter $\beta$ in $(0,1]$, the posterior concentration rate is $n^{-\frac{\beta}{2\beta+p}}(\log n)^{2+\frac{p}{2\beta}}$, whereas the minimax rate for one-dimensional Hölder continuous functions is $(n/\log n)^{-\beta/(2\beta+1)}$.

- When the true density function is sparse in the sense that the Haar wavelet coefficients satisfy a weak-$l_q$ ($q > 1/2$) constraint, the posterior concentration rate is $n^{-\frac{q-1/2}{2q}}(\log n)^{2+\frac{1}{2q-1}}$.

- We can use a computationally efficient algorithm to sample from the posterior distribution. We demonstrate the theoretical results on several simulated data sets.

## 1.1 Related work

An important feature of our method is that it can adapt to the unknown smoothness of the true density function. The adaptivity of Bayesian approaches has drawn great attention in recent years. In terms of density estimation, there are mainly two categories of adaptive Bayesian nonparametric approaches. The first category of work relies on basis expansion of the density function and typically imposes a random series prior [15, 17]. When the prior on the coefficients of the expansion is set to be normal [4], it is also a Gaussian process prior. In the multivariate case, most existing work [4, 17] uses tensor-product basis. Our improvement over these methods mainly lies in the adaptive structure. In fact, as the dimension increases the number of tensor-product basis functions can be prohibitively large, which imposes a great challenge on computation. By introducing adaptive partition, we are able to handle the multivariate case even when the dimension is 30 (Example 2 in Section 4).

Another line of work considers mixture priors [16, 11, 18]. Although the mixture distributions have good approximation properties and naturally lead to adaptivity to very high smoothness levels, they may fail to detect or characterize the local features. On the other hand, by learning a partition of the sample space, the partition based approaches can provide an informative summary of the structure, and allow us to examine the density at different resolutions [14, 21].

The paper is organized as follows. In Section 2 we provide more details of the density functions on binary partitions and define the prior distribution. Section 3 summarizes the theoretical results on posterior concentration rates. The results are further validated in Section 4 by several experiments.

## 2 Bayesian multivariate density estimation

We focus on density estimation problems in $p$-dimensional Euclidean space. Let $(\Omega, \mathcal{B})$ be a measurable space and $f_0$ be a compactly supported density function with respect to the Lebesgue measure $\mu$. $Y_1, Y_2, \cdots, Y_n$ is a sequence of independent variables distributed according to $f_0$. After translation and scaling, we can always assume that the support of $f_0$ is contained in the unit cube in $\mathbb{R}^p$. Translating this into notations, we assume that $\Omega = \{(y^1, y^2, \cdots, y^p) : y^l \in [0,1]\}$. $\mathcal{F} = \{f \text{ is a nonnegative measurable function on } \Omega : \int_\Omega f d\mu = 1\}$ denotes the collection of all the density functions on $(\Omega, \mathcal{B}, \mu)$. Then $\mathcal{F}$ constitutes the parameter space in this problem. Note that $\mathcal{F}$ is an infinite dimensional parameter space.

## 2.1 Densities on binary partitions

To address the infinite dimensionality of $\mathcal{F}$, we construct a sequence of finite dimensional approximating spaces $\Theta_1, \Theta_2, \cdots, \Theta_I, \cdots$ based on *binary partitions*. With growing complexity, these spaces provide more and more accurate approximations to the initial parameter space $\mathcal{F}$. Here, we use a recursive procedure to define a binary partition with $I$ subregions of the unit cube in $\mathbb{R}^p$. Let $\Omega = \{(y^1, y^2, \cdots, y^p) : y^l \in [0,1]\}$ be the unit cube in $\mathbb{R}^p$. In the first step, we choose one of the coordinates $y^l$ and cut $\Omega$ into two subregions along the midpoint of the range of $y^l$. That is, $\Omega = \Omega_0^l \cup \Omega_1^l$, where $\Omega_0^l = \{y \in \Omega : y^l \le 1/2\}$ and $\Omega_1^l = \Omega \backslash \Omega_0^l$. In this way, we get a partition

with two subregions. Note that the total number of possible partitions after the first step is equal to the dimension $p$. Suppose after $I-1$ steps of the recursion, we have obtained a partition $\{\Omega_i\}_{i=1}^{I}$ with $I$ subregions. In the $I$-th step, further partitioning of the region is defined as follows:

1. Choose a region from $\Omega_1, \cdots, \Omega_I$. Denote it as $\Omega_{i_0}$.

2. Choose one coordinate $y^l$ and divide $\Omega_{i_0}$ into two subregions along the midpoint of the range of $y^l$.

Such a partition obtained by $I-1$ recursive steps is called a binary partition of size $I$. Figure 1 displays all possible two dimensional binary partitions when $I$ is 1, 2 and 3.

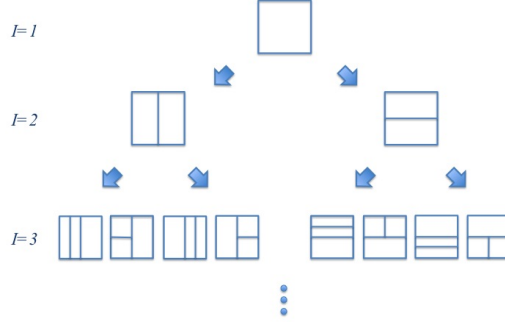

Figure 1: Binary partitions

Now, let

$$\Theta_I = \{f : f = \sum_{i=1}^{I} \frac{\theta_i}{|\Omega_i|} \mathbb{1}_{\Omega_i}, \sum_{i=1}^{I} \theta_i = 1, \ \{\Omega_i\}_{i=1}^{I} \text{ is a binary partition of } \Omega.\}$$

where $|\Omega_i|$ is the volume of $\Omega_i$. Then, $\Theta_I$ is the collection of the density functions supported by the binary partitions of size $I$. They constitute a sequence of approximating spaces (i.e. a sieve, see [10, 20] for background on sieve theory). Let $\Theta = \cup_{I=1}^{\infty} \Theta_I$ be the space containing all the density functions supported by the binary partitions. Then $\Theta$ is an approximation of the initial parameter space $\mathcal{F}$ to certain approximation error which will be characterized later.

We take the metric on $\mathcal{F}$, $\Theta$ and $\Theta_I$ to be Hellinger distance, which is defined as

$$\rho(f,g) = (\int_{\Omega} (\sqrt{f(y)} - \sqrt{g(y)})^2 dy)^{1/2}, \ f,g \in \mathcal{F}. \tag{1}$$

## 2.2 Prior distribution

An ideal prior $\Pi$ on $\Theta = \cup_{I=1}^{\infty} \Theta_I$ is supposed to be capable of balancing the approximation error and the complexity of $\Theta$. The prior in this paper penalizes the size of the partition in the sense that the probability mass on each $\Theta_I$ is proportional to $\exp(-\lambda I \log I)$. Given a sample of size $n$, we restrict our attention to $\Theta_n = \cup_{I=1}^{n/\log n} \Theta_I$, because in practice we need enough samples within each subregion to get a meaningful estimate of the density. This is to say, when $I \leq n/\log n$, $\Pi(\Theta_I) \propto \exp(-\lambda I \log I)$, otherwise $\Pi(\Theta_I) = 0$.

If we use $T_I$ to denote the total number of possible partitions of size $I$, then it is not hard to see that $\log T_I \leq c^* I \log I$, where $c^*$ is a constant. Within each $\Theta_I$, the prior is uniform across all binary partitions. In other words, let $\{\Omega_i\}_{i=1}^{I}$ be a binary partition of $\Omega$ of size $I$, and $\mathcal{F}(\{\Omega_i\}_{i=1}^{I})$ is the collection of piecewise constant density functions on this partition (i.e. $\mathcal{F}(\{\Omega_i\}_{i=1}^{I}) = \{f = \sum_{i=1}^{I} \frac{\theta_i}{|\Omega_i|} \mathbb{1}_{\Omega_i} : \sum_{i=1}^{I} \theta_i = 1 \text{ and } \theta_i \geq 0, i = 1, \ldots, I\}$), then

$$\Pi\left(\mathcal{F}\left(\{\Omega_i\}_{i=1}^{I}\right)\right) \propto \exp(-\lambda I \log I)/T_I. \tag{2}$$

Given a partition $\{\Omega_i\}_{i=1}^I$, the weights $\theta_i$ on the subregions follow a Dirichlet distribution with parameters all equal to $\alpha$ ($\alpha < 1$). This is to say, for $x_1, \cdots, x_I \geq 0$ and $\sum_{i=1}^I x_i = 1$,

$$\Pi\left(f = \sum_{i=1}^I \frac{\theta_i}{|\Omega_i|} \mathbb{1}_{\Omega_i} : \theta_1 \in dx_1, \cdots, \theta_I \in dx_I \middle| \mathcal{F}\left(\{\Omega_i\}_{i=1}^I\right)\right) = \frac{1}{D(\alpha, \cdots, \alpha)} \prod_{i=1}^I x_i^{\alpha-1}, \quad (3)$$

where $D(\delta_1, \cdots, \delta_I) = \prod_{i=1}^I \Gamma(\delta_i)/\Gamma(\sum_{i=1}^I \delta_i)$.

Let $\Pi_n(\cdot|Y_1, \cdots, Y_n)$ to denote the posterior distribution. After integrating out the weights $\theta_i$, we can compute the marginal posterior probability of $\mathcal{F}\left(\{\Omega_i\}_{i=1}^I\right)$:

$$
\begin{aligned}
\Pi_n\left(\mathcal{F}(\{\Omega_i\}_{i=1}^I)\middle|Y_1, \cdots, Y_n\right) &\propto \Pi\left(\mathcal{F}(\{\Omega_i\}_{i=1}^I)\right) \int \left(\prod_{i=1}^I (\theta_i/|\Omega_i|)^{n_i}\right) \\
&\quad \times \left(\frac{1}{D(\alpha, \cdots, \alpha)} \prod_{i=1}^I \theta_i^{\alpha-1}\right) d\theta_1 \cdots d\theta_I \\
&\propto \frac{\exp(-\lambda I \log I)}{T_I} \cdot \frac{D(\alpha + n_1, \cdots, \alpha + n_I)}{D(\alpha, \cdots, \alpha)} \prod_{i=1}^I \frac{1}{|\Omega_i|^{n_i}}, \quad (4)
\end{aligned}
$$

where $n_i$ is the number of observations in $\Omega_i$. Under the prior introduced in [13], the marginal posterior distribution is:

$$\Pi_n^*\left(\mathcal{F}(\{\Omega_i\}_{i=1}^I)\middle|Y_1, \cdots, Y_n\right) \propto \exp(-\lambda I) \frac{D(\alpha + n_1, \cdots, \alpha + n_I)}{D(\alpha, \cdots, \alpha)} \prod_{i=1}^I \frac{1}{|\Omega_i|^{n_i}}, \quad (5)$$

while the maximum log-likelihood achieved by histograms on the partition $\{\Omega_i\}_{i=1}^n$ is:

$$l_n(\mathcal{F}(\{\Omega_i\}_{i=1}^I)) := \max_{f \in \mathcal{F}(\{\Omega_i\}_{i=1}^I)} l_n(f) = \sum_{i=1}^I n_i \log\left(\frac{n_i}{n|\Omega_i|}\right). \quad (6)$$

From a model selection perspective, we may treat the histograms on each binary partition as a model of the data. When $I \ll n$, asymptotically,

$$\log\left(\Pi_n^*\left(\mathcal{F}(\{\Omega_i\}_{i=1}^I)\middle|Y_1, \cdots, Y_n\right)\right) \asymp l_n(\mathcal{F}(\{\Omega_i\}_{i=1}^I)) - \frac{1}{2}(I-1)\log n. \quad (7)$$

This is to say, in [13], selecting the partition which maximizes the marginal posterior distribution is equivalent to applying the Bayesian information criterion (BIC) to perform model selection. However, if we allow $I$ to increase with $n$, (7) will not hold any more. But if we use the prior introduced in this section, in the case when $\frac{I}{n} \to \zeta \in (0,1)$ as $n \to \infty$, we still have

$$\log\left(\Pi_n\left(\mathcal{F}(\{\Omega_i\}_{i=1}^I)\middle|Y_1, \cdots, Y_n\right)\right) \asymp l_n(\mathcal{F}(\{\Omega_i\}_{i=1}^I)) - \lambda I \log I. \quad (8)$$

From a model selection perspective, this is closer to the risk inflation criterion (RIC, [8]).

## 3  Posterior concentration rates

We are interested in how fast the posterior probability measure concentrates around the true the density $f_0$. Under the prior specified above, the posterior probability is the random measure given by

$$\Pi_n(B|Y_1, \cdots, Y_n) = \frac{\int_B \prod_{j=1}^n f(Y_j) d\Pi(f)}{\int_\Theta \prod_{j=1}^n f(Y_j) d\Pi(f)}.$$

A Bayesian estimator is said to be *consistent* if the posterior distribution concentrates on arbitrarily small neighborhoods of $f_0$, with probability tending to 1 under $P_0^n$ ($P_0$ is the probability measure corresponding to the density function $f_0$). The posterior concentration rate refers to the rate at which these neighborhoods shrink to zero while still possessing most of the posterior mass. More explicitly, we want to find a sequence $\epsilon_n \to 0$, such that for sufficiently large $M$,

$$\Pi_n\left(\{f : \rho(f, f_0) \geq M\epsilon_n\}|Y_1, \cdots, Y_n\right) \to 0 \text{ in } P_0^n - \text{probability}.$$

In [6] and [2], the authors demonstrated that it is impossible to find an estimator which works uniformly well for every $f$ in $\mathcal{F}$. This is the case because for any estimator $\hat{f}$, there always exists $f \in \mathcal{F}$ for which $\hat{f}$ is inconsistent. Given the minimaxity of the Bayes estimator, we have to restrict our attention to a subset of the original parameter space $\mathcal{F}$. Here, we focus on the class of density functions that can be well approximated by $\Theta_I$'s. To be more rigorous, a density function $f \in \mathcal{F}$ is said to be well approximated by elements in $\Theta$, if there exits a sequence of $f_I \in \Theta_I$, satisfying that $\rho(f_I, f) = O(I^{-r})(r > 0)$. Let $\mathcal{F}_0$ be the collection of these density functions. We will first derive posterior concentration rate for the elements in $\mathcal{F}_0$ as a function of $r$. For different function classes, this approximation rate $r$ can be calculated explicitly. In addition to this, we also assume that $f_0$ has finite second moment.

The following theorem gives the posterior concentration rate under the prior introduced in Section 2.2.

**Theorem 3.1.** *$Y_1, \cdots, Y_n$ is a sequence of independent random variables distributed according to $f_0$. $P_0$ is the probability measure corresponding to $f_0$. $\Theta$ is the collection of $p$-dimensional density functions supported by the binary partitions as defined in Section 2.1. With the modified prior distribution, if $f_0 \in \mathcal{F}_0$, then the posterior concentration rate is $\epsilon_n = n^{-\frac{r}{2r+1}}(\log n)^{2+\frac{1}{2r}}$.*

The strategy to show this theorem is to write the posterior probability of the shrinking ball as

$$\Pi(\{f : \rho(f, f_0) \geq M\epsilon_n\}|Y_1, \cdots, Y_n) = \frac{\sum_{I=1}^{\infty} \int_{\{f:\rho(f,f_0)\geq M\epsilon_n\}\cap\Theta_I} \prod_{j=1}^{n} \frac{f(Y_j)}{f_0(Y_j)} d\Pi(f)}{\sum_{I=1}^{\infty} \int_{\Theta_I} \prod_{j=1}^{n} \frac{f(Y_j)}{f_0(Y_j)} d\Pi(f)}. \tag{9}$$

The proof employs the mechanism developed in the landmark works [9] and [19]. We first obtain the upper bounds for the items in the numerator by dividing them into three blocks, each of which accounts for bias, variance, and rapidly decaying prior respectively, and calculate the upper bound for each block separately. Then we provide the prior thickness result, i.e., we bound the prior mass of a ball around the true density from below. Due to space constraints, the details of the proof will be provided in the appendix.

This theorem suggests the following two take-away messages: 1. The rate is adaptive to the unknown smoothness of the true density. 2. The posterior contraction rate is $n^{-\frac{r}{2r+1}}(\log n)^{2+\frac{1}{2r}}$, which does not directly depend on the dimension $p$. For some density functions, $r$ may depend on $p$. But in several special cases, like the density function is spatially sparse or the density function lies in a low dimensional subspace, we will show that the rate will not be affected by the full dimension of the problem.

In the following three subsections, we will calculate the explicit rates for three density classes. Again, all proofs are given in the appendix.

### 3.1 Spatial adaptation

First, we assume that the density concentrates spatially. Mathematically, this implies the density function satisfies a type of *sparsity*. In the past two decades, sparsity has become one of the most discussed types of structure under which we are able to overcome the curse of dimensionality. A remarkable example is that it allows us to solve high-dimensional linear models, especially when the system is underdetermined.

Let $f$ be a $p$ dimensional density function and $\Psi$ the $p$-dimensional Haar basis. We will work with $g = \sqrt{f}$ first. Note that $g \in L^2([0,1]^p)$. Thus we can expand $g$ with respect to $\Psi$ as $g = \sum_{\psi \in \Psi} \langle g, \psi \rangle \psi$, $\psi \in \Psi$. We rearrange this summation by the size of wavelet coefficients. In other words, we order the coefficients as the following

$$|\langle g, \psi_{(1)} \rangle| \geq |\langle g, \psi_{(2)} \rangle| \geq \cdots \geq |\langle g, \psi_{(k)} \rangle| \geq \cdots,$$

then the sparsity condition imposed on the density functions is that the decay of the wavelet coefficients follows a power law,

$$|\langle g, \psi_{(k)} \rangle| \leq Ck^{-q} \text{ for all } k \in \mathbb{N} \text{ and } q > 1/2, \tag{10}$$

where $C$ is a constant.

We call such a constraint a weak-$l_q$ constraint. The condition has been widely used to characterize the sparsity of signals and images [1, 3]. In particular, in [5], it was shown that for two-dimensional cases, when $q > 1/2$, this condition reasonably captures the sparsity of real world images.

**Corollary 3.2.** *(Application to spatial adaptation) Suppose $f_0$ is a p-dimensional density function and satisfies the condition (10). If we apply our approaches to this type of density functions, the posterior concentration rate is $n^{-\frac{q-1/2}{2q}}(\log n)^{2+\frac{1}{2q-1}}$.*

## 3.2 Density functions of bounded variation

Let $\Omega = [0, 1)^2$ be a domain in $\mathbb{R}^2$. We first characterize the space $BV(\Omega)$ of functions of bounded variation on $\Omega$.

For a vector $\nu \in \mathbb{R}^2$, the difference operator $\Delta_\nu$ along the direction $\nu$ is defined by

$$\Delta_\nu(f, y) := f(y + \nu) - f(y).$$

For functions $f$ defined on $\Omega$, $\Delta_\nu(f, y)$ is defined whenever $y \in \Omega(\nu)$, where $\Omega(\nu) := \{y : [y, y + \nu] \subset \Omega\}$ and $[y, y + \nu]$ is the line segment connecting $y$ and $y + \nu$. Denote by $e_l, l = 1, 2$ the two coordinate vectors in $\mathbb{R}^2$. We say that a function $f \in L_1(\Omega)$ is in $BV(\Omega)$ if and only if

$$V_\Omega(f) := \sup_{h>0} h^{-1} \sum_{l=1}^{2} \|\Delta_{he_l}(f, \cdot)\|_{L_1(\Omega(he_l))} = \lim_{h \to 0} h^{-1} \sum_{l=1}^{2} \|\Delta_{he_l}(f, \cdot)\|_{L_1(\Omega(he_l))}$$

is finite. The quantity $V_\Omega(f)$ is the *variation* of $f$ over $\Omega$.

**Corollary 3.3.** *Assume that $f_0 \in BV(\Omega)$. If we apply the Bayesian multivariate density estimator based on adaptive partitioning here to estimate $f_0$, the posterior concentration rate is $n^{-1/4}(\log n)^3$.*

## 3.3 Hölder space

In one-dimensional case, the class of Hölder functions $\mathcal{H}(L, \beta)$ with regularity parameter $\beta$ is defined as the following: let $\kappa$ be the largest integer smaller than $\beta$, and denote by $f^{(\kappa)}$ its $\kappa$th derivative.

$$\mathcal{H}(L, \beta) = \{f : [0, 1] \to \mathbb{R} : |f^{(\kappa)}(x) - f^{(\kappa)}(y)| \leq L|x - y|^{\beta - \kappa}\}.$$

In multi-dimensional cases, we introduce the Mixed-Hölder continuity. In order to simplify the notation, we give the definition when the dimension is two. It can be easily generalized to high-dimensional cases. A real-valued function $f$ on $\mathbb{R}^2$ is called Mixed-Hölder continuous for some nonnegative constant $C$ and $\beta \in (0, 1]$, if for any $(x_1, y_1), (x_1, y_2) \in \mathbb{R}^2$,

$$|f(x_2, y_2) - f(x_2, y_1) - f(x_1, y_2) + f(x_1, y_1)| \leq C|x_1 - x_2|^\beta |y_1 - y_2|^\beta.$$

**Corollary 3.4.** *Let $f_0$ be the p-dimensional density function. If $\sqrt{f_0}$ is Hölder continuous (when $p = 1$) or mixed-Hölder continuous (when $p \geq 2$) with regularity parameter $\beta \in (0, 1]$, then the posterior concentration rate of the Bayes estimator is $n^{-\frac{\beta}{2\beta+p}}(\log n)^{2+\frac{p}{2\beta}}$.*

This result also implies that if $f_0$ only depends on $\tilde{p}$ variable where $\tilde{p} < p$, but we do not know in advance which $\tilde{p}$ variables, then the rate of this method is determined by the effective dimension $\tilde{p}$ of the problem, since the smoothness parameter $r$ is only a function of $\tilde{p}$. In next section, we will use a simulated data set to illustrate this point.

# 4 Simulation

## 4.1 Sequential importance sampling

Each partition $\mathcal{A}_I = \{\Omega_i\}_{i=1}^{I}$ is obtained by recursively partitioning the sample space. We can use a sequence of partitions $\mathcal{A}_1, \mathcal{A}_2, \cdots, \mathcal{A}_I$ to keep track of the path leading to $\mathcal{A}_I$. Let $\Pi_n(\cdot)$ denote the posterior distribution $\Pi_n(\cdot|Y_1, \cdots, Y_n)$ for simplicity, and $\Pi_n^I$ be the posterior distribution conditioning on $\Theta_I$. Then $\Pi_n^I(\mathcal{A}_I)$ can be decomposed as

$$\Pi_n^I(\mathcal{A}_I) = \Pi_n^I(\mathcal{A}_1)\Pi_n^I(\mathcal{A}_2|\mathcal{A}_1)\cdots\Pi_n^I(A_I|A_{I-1}).$$

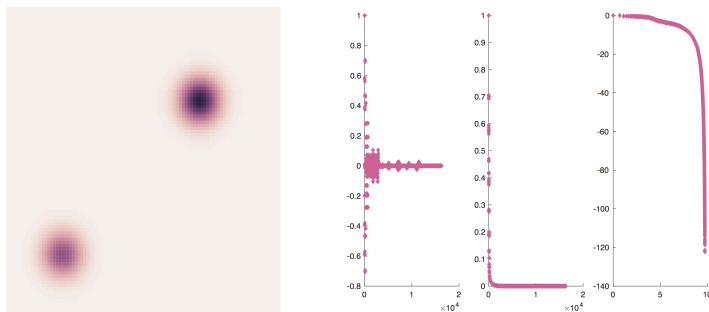

Figure 2: Heatmap of the density and plots of the 2-dimensional Haar coefficients. For the plot on the right, the left panel is the plot of the Haar coefficients from low resolution to high resolution up to level 6. The middle one is the plot of the sorted coefficients according to their absolute values. And the right one is the same as the middle plot but with the abscissa in log scale.

The conditional distribution $\Pi_n^I(\mathcal{A}_{i+1}|\mathcal{A}_i)$ can be calculated by $\Pi_n^I(\mathcal{A}_{i+1})/\Pi_n^I(\mathcal{A}_i)$. However, the computation of the marginal distribution $\Pi_n^I(\mathcal{A}_i)$ is sometimes infeasible, especially when both $I$ and $I-i$ are large, because we need to sum the marginal posterior probability over all binary partitions of size $I$ for which the first $i$ steps in the partition generating path are the same as those of $\mathcal{A}_i$. Therefore, we adopt the sequential importance algorithm proposed in [13]. In order to build a sequence of binary partitions, at each step, the conditional distribution is approximated by $\Pi_n^{i+1}(\mathcal{A}_{i+1}|\mathcal{A}_i)$. The obtained partition is assigned a weight to compensate the approximation, where the weight is

$$w_I(\mathcal{A}_I) = \frac{\Pi_n^I(\mathcal{A}_I)}{\Pi_n^1(\mathcal{A}_1)\Pi_n^2(\mathcal{A}_2|\mathcal{A}_1)\cdots\Pi_n^I(\mathcal{A}_I|\mathcal{A}_{I-1})}.$$

In order to make the data points as uniform as possible, we apply a copula transformation to each variable in advance whenever the dimension exceeds 3. More specifically, we estimate the marginal distribution of each variable $X_j$ by our approach, denoted as $\hat{f}_j$ (we use $\hat{F}_j$ to denote the cdf of $X_j$), and transform each point $(y^1, \cdots, y^p)$ to $(F_1(y^1), \cdots, F_p(y^p))$. Another advantage of this transformation is that after the transformation the sample space naturally becomes $[0, 1]^p$.

**Example 1**  Assume that the two-dimensional density function is

$$\binom{Y_1}{Y_2} \sim \frac{2}{5}\mathcal{N}\left(\binom{0.25}{0.25}, 0.05^2 I_{2\times2}\right) + \frac{3}{5}\mathcal{N}\left(\binom{0.75}{0.75}, 0.05^2 I_{2\times2}\right).$$

This density function both satisfies the spatial sparsity condition and belongs to the space of functions of bounded variation. Figure 2 shows the heatmap of the density function and its Haar coefficients. The last panel in the second plot displays the sorted coefficients with the abscissa in log-scale. From this we can clearly see that the power-law decay defined in Section 3.1 is satisfied.

We apply the adaptive partitioning approach to estimate the density, and allow the sample size increase from $10^2$ to $10^5$. In Figure 3, the left plot is the density estimation result based on a sample with 10000 data points. The right one is the plot of Kullback-Leibler (KL) divergence from the estimated density to $f_0$ vs. sample size in log-scale. The sample sizes are set to be 100, 500, 1000, 5000, $10^4$, and $10^5$. The linear trend in the plot validates the posterior concentrate rates calculated in Section 3. The reason why we use KL divergence instead of the Hellinger distance is that for any $f_0 \in \mathcal{F}_0$ and $\hat{f} \in \Theta$, we can show that the KL divergence and the Hellinger distance are of the same order. But KL divergence is relatively easier to compute in our setting, since we can show that it is linear in the logarithm of the posterior marginal probability of a partition. The proof will be provided in the appendix. For each fixed sample size, we run the experiment 10 times and estimate the standard error, which is shown by the lighter blue part in the plot.

**Example 2**  In the second example we work with a density function of moderately high dimension. Assume that the first five random variables $Y_1, \cdots Y_5$ are generated from the following location

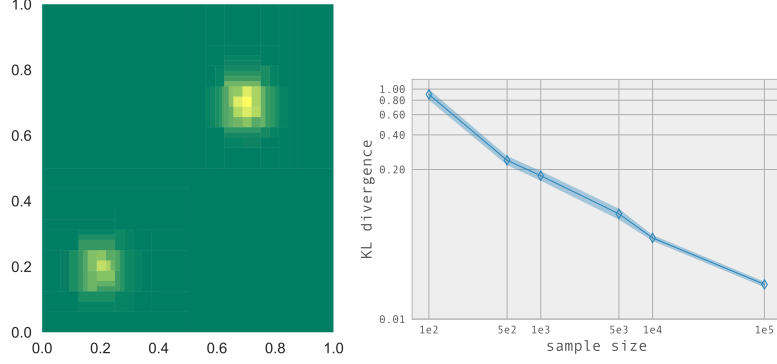

Figure 3: Plot of the estimated density and KL divergence against sample size. We use the posterior mean as the estimate. The right plot is on log-log scale, while the labels of $x$ and $y$ axes still represent the sample size and the KL divergence before we take the logarithm.

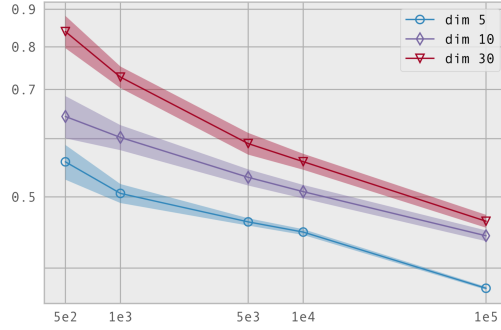

Figure 4: KL divergence vs. sample size. The blue, purple and red curves correspond to the cases when $p = 5$, $p = 10$ and $p = 30$ respectively. The slopes of the three lines are almost the same, implying that the concentration rate only depends on the effective dimension of the problem (which is 5 in this example).

mixture of the Gaussian distribution:

$$
\begin{pmatrix} Y_1 \\ Y_2 \\ Y_3 \end{pmatrix} \sim \frac{1}{2}\mathcal{N}\left( \begin{pmatrix} 0.25 \\ 0.25 \\ 0.25 \end{pmatrix}, \begin{pmatrix} 0.05^2 & 0.03^2 & 0 \\ 0.03^2 & 0.05^2 & 0 \\ 0 & 0 & 0.05^2 \end{pmatrix} \right) + \frac{1}{2}\mathcal{N}\left( \begin{pmatrix} 0.75 \\ 0.75 \\ 0.75 \end{pmatrix}, 0.05^2 I_{3\times 3} \right),
$$

$$
Y_4, Y_5 \sim \mathcal{N}(0.5, 0.1),
$$

the other components $Y_6, \cdots, Y_p$ are independently uniformly distributed. We run experiments for $p = 5, 10$, and $30$. For a fixed $p$, we generate $n \in \{500, 1000, 5000, 10^4, 10^5\}$ data points. For each pair of $p$ and $n$, we repeat the experiment 10 times and calculate the standard error. Figure 4 displays the plot of the KL divergence vs. the sample size on log-log scale. The density function is continuous differentiable. Therefore, it satisfies the mixed-Hölder continuity condition. The effective dimension of this example is $\tilde{p} = 5$, and this is reflected in the plot: the slopes of the three lines, which correspond to the concentration rates under different dimensions, almost remain the same as we increase the full dimension of the problem.

## 5 Conclusion

In this paper, we study the posterior concentration rate of a class of Bayesian density estimators based on adaptive partitioning. We obtain explicit rates when the density function is spatially sparse, belongs to the space of bounded variation, or is Hölder continuous. For the last case, the rate is minimax up to a logarithmic term. When the density function is sparse or lies in a low-dimensional subspace, the rate will not be affected by the dimension of the problem. Another advantage of this method is that it can adapt to the unknown smoothness of the underlying density function.

## Footnotes

*Work was done while the author was at Stanford University.

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
