[Supplementary Material]

# Supplementary Material

**Linxi Liu**
Columbia University
New York, NY 10027
ll3098@columbia.edu

**Dangna Li**
Stanford University
Stanford, CA 94305
dangna@stanford.edu

**Wing Hung Wong**
Stanford University
Stanford, CA 94305
whwong@stanford.edu

## A  Upper bound of the numerator

Briefly speaking, the numerator can be bounded by controlling the complexity of the parameter space $\Theta$. Here, the complexity of the model is measured by the *metric entropy*. A general discussion of metric entropy can be found in [4]. In this section, we introduce a form of metric entropy with bracketing corresponding to the relavent parameter space, and provide an upper bound for the metric entropy of the approximating spaces defined in Section 2. These bounds lead to upper bounds for the items in the numerator of (9).

**Definition A.1.** *Let $(\Theta, \rho)$ be a seperable pseudo-metric space. $\Theta(\epsilon)$ is a finite set of pairs of functions $\{(f_j^L, f_j^U), j = 1, \cdots, N\}$ satisfying*

$$\rho(f_j^L, f_j^U) \le \epsilon \ for \ j = 1, \cdots, N, \tag{11}$$

*and for any $f \in \Theta$, there is a j such that*

$$f_j^L \le f \le f_j^U. \tag{12}$$

*Let*

$$N(\epsilon, \Theta, \rho) = min\{card\ \Theta(\epsilon) : (11)\ and\ (12)\ are\ satisfied\}. \tag{13}$$

*Then, we define the metric entropy with bracketing of $\Theta$ to be*

$$H(\epsilon, \Theta, \rho) = \log N(\epsilon, \Theta, \rho). \tag{14}$$

Recall that $\Theta_1, \cdots, \Theta_I, \cdots$ are the approximating spaces defined in Section 2. The next lemma is devoted to an upper bound for the bracketing metric entropy of $\Theta_I$.

**Lemma A.2.** *Take $\rho$ to be the Hellinger distance. Let $\Theta_I^d = \{f \in \Theta_I : \rho(f, f_0) \le d\}$. Then,*

$$H(\epsilon, \Theta_I^d, \rho)$$
$$\le \quad I \log p + (I+1) \log(I+1) + \frac{I}{2} \log I + I \log \frac{d}{\epsilon} + c', \tag{15}$$

*where c is a constant not dependent on I or d.*

*Proof.* See [5] proof of Lemma 3.1 and Lemma 3.2. □

Our next theorem, which is Theorem 1 in [7], gives a uniform exponential bound for likelihood ratios.

**Theorem A.3** (Wong and Shen (1995))**.** *There exist positive constants $a > 0$, $c$, $c_1$ and $c_2$, such that, for any $\epsilon > 0$, if*

$$\int_{\epsilon^2/8}^{\sqrt{2}\epsilon} H^{1/2}(u/a, \mathcal{P}, \rho) du \le cn^{1/2}\epsilon^2, \tag{16}$$

*then*

$$\mathbb{P}_{f_0}\Big( \sup_{\{\rho(f, f_0) \ge \epsilon, f \in \mathcal{P}\}} \prod_{i=1}^{n} \frac{f(Y_i)}{f_0(Y_i)} \ge \exp(-c_1 n\epsilon^2) \Big) \le 4 \exp(-c_2 n\epsilon^2),$$

*where $\mathbb{P}_{f_0}$ is understood to be the outer probability mesure under $f_0$. The constants $c_1$ and $c_2$ can be chosen in $(0, 1)$ and $c$ can be set as $(2/3)^{5/2}/512$.*

Finally, the next lemma provides an upper bound for the items in the numerator in (9) when $I$ is sufficiently large.

**Lemma A.4.** *Let* $\delta_{n,I} = (\frac{I \log I}{n/\log n})^{1/2}$. *When $n$ is sufficiently large, we have*

$$\mathbb{P}_{f_0}\left(\sup_{\{\rho(f,f_0) \geq \delta_{n,I}, f \in \Theta_I\}} \prod_{i=1}^n \frac{f(Y_i)}{f_0(Y_i)} \geq \exp(-c_1 n \delta_{n,I}^2)\right) \leq 4\exp(-c_2 n \delta_{n,I}^2).$$

*Proof.* See [5] proof of Corollary 3.1. $\square$

**Remark A.1.** *Since the metric entropy decreases as $\epsilon$ increases, this lemma also holds for any $\epsilon \geq \delta_{n,I}$. This property is quite useful in the proof of the main theorem.*

# B    Lower bound of the denominator

In this section, we study how the prior distribution concentrates on the shrinking neighborhoods around the true density function. This is the key to bounding the denominator of (9) from below. We develop our results through a series of lemmas. The connection between the lower bounds of the items in the denominator of (9) and the concentration rate of the prior distribution is first derived (B.1). By employing a property of Dirichlet distribution (Lemma B.3) and inequalities bounding Kullback-Leibler divergence by Hellinger distance (Lemma B.2), we obtain lower bounds of the terms in the denominator of (9) in Lemma B.4.

To begin with, we cite a result from [6]. In this lemma, it is shown that with probability close to 1, the denominator is bounded from below by the prior probability mass concentrating on a ball around $f_0$ multiplied by a coefficient depending on the radius of the ball. Before stating the result, we define

$$K(f_0, f) = \mathbb{E}_{f_0}\left(\log \frac{f_0(Y)}{f(Y)}\right), \tag{17}$$

and

$$V(f_0, f) = \text{Var}_{f_0}\left(\log \frac{f_0(Y)}{f(Y)}\right). \tag{18}$$

**Lemma B.1** (Shen and Wasserman (2001) Lemma 1)**.** *Let $K(\cdot, \cdot)$ and $V(\cdot, \cdot)$ be as defined in (17) and (18), and let $S(t) = \{f \in \Omega : K(f_0, f) \leq t, V(f_0, f) \leq t\}$. Set $S_n = S(t_n)$. When $t_n$ is a sequence of positive numbers satisfying $nt_n \to \infty$,*

$$\mathbb{P}_{f_0}^n\left(\int_\Omega \prod_{j=1}^n \frac{f(Y_i)}{f_0(Y_i)} d\Pi(f) \leq \frac{1}{2}\Pi(S_n)e^{-2nt_n}\right) \leq \frac{2}{nt_n}.$$

More explicitly, from this lemma we learn that, given the condition $nt_n \to \infty$, $\int_\Omega \prod_{j=1}^n \frac{f(Y_i)}{f_0(Y_i)} d\Pi(f) \geq \frac{1}{2}\Pi(S_n)e^{-2nt_n}$ with probability close to 1.

It is well known that Hellinger distance can be bounded by the Kullback-Leibler divergence. In [7], it is shown that the other direction also holds under an integrability condition. Their results are summarized in the lemma below.

**Lemma B.2** (Wong and Shen (1995) Theorem 5)**.** *Let $f$, $f_0$ be two densities, $\rho^2(f, f_0) \leq \epsilon^2$. Suppose that $M_\delta^2 = \int_{\{f_0/f \geq e^{1/\delta}\}} f_0(f_0/f)^\delta < \infty$ for some $\delta \in (0, 1]$. Then for all $\epsilon^2 \leq \frac{1}{2}(1 - e^{-1})^2$, we have*

$$\int f_0 \log(\frac{f_0}{f}) \leq \left[6 + \frac{2 \log 2}{(1 - e^{-1})^2} + \frac{8}{\delta}\max\left(1, \log(\frac{M_\delta}{\epsilon})\right)\right]\epsilon^2,$$

$$\int f_0\left(\log(\frac{f_0}{f})\right)^2 \leq 5\epsilon^2\left[\frac{1}{\delta}\max\left(1, \log(\frac{M_\delta}{\epsilon})\right)\right]^2.$$

Based on this result, if $\rho^2(f, f_0) \leq \epsilon^2$, then

$$\max\left(K(f_0, f), \mathbb{E}_{f_0}\left((\log \frac{f_0(Y)}{f(Y)})^2\right)\right) = O\left(\epsilon^2(\log \frac{M_\delta}{\epsilon})^2\right).$$

This further implies that, there exists a constant $L$, such that

$$\left\{ f : \rho(f, f_0) \leq \frac{L\epsilon}{\log \frac{M_\delta}{\epsilon}} \right\}$$

$$\subset \quad \left\{ f : K(f_0, f) \leq \epsilon^2, \mathbb{E}_{f_0}\big((\log \frac{f_0(Y)}{f(Y)})^2\big) \leq \epsilon^2 \right\}. \tag{19}$$

This lemma allows us to work on a Hellinger ball instead of a Kullback-Leibler one. The transition is necessary because it is more straightforward to apply a property of the Dirichlet distribution to estimate the probability mass on a Hellinger ball around the true density function. In the lemma below, this particular property of the Dirichlet distribution is stated in terms of $L_1$ distance, which is equivalent to the Hellinger distance. We want to point out that this lemma is a variation of Lemma 6.1 in [2] and the proof is adapted from their paper.

**Lemma B.3.** $(X_1, \cdots, X_I)$ *is distributed according to the Dirichlet distribution. Let* $(x_{10}, \cdots, x_{I0})$ *be any point on the $I$-simplex. Take $\epsilon < 1/I$. With $\tau < \epsilon^2$, we have*

$$P\Big( \sum_{i=1}^{I} |X_i - x_{i0}| \leq 2\epsilon, X_i \geq \tau \text{ for all } i \Big) \geq \frac{\Gamma(\alpha I)}{(\Gamma(\alpha))^I} (\epsilon^2 - \tau)^I. \tag{20}$$

*Proof.* We can find an index $i$ such that $x_{i0} > 1/I$. By relabeling, we can assume that $i = I$. if $|x_i - x_{i0}| \leq \epsilon^2$ for $i = 1, \cdots, I - 1$, then

$$\sum_{i=1}^{I-1} x_i \leq 1 - x_{I0} + (I-1)\epsilon^2 \leq (I-1)(\epsilon^2 + 1/I) \leq 1 - \epsilon^2 < 1.$$

Therefore, there exists $x = (x_1, \cdots, x_I)$ in the simplex with these first $I - 1$ coordinates. And

$$\sum_{i=1}^{I} |x_i - x_{i0}| \leq 2 \sum_{i=1}^{I-1} |x_i - x_{i0}| \leq 2\epsilon^2 (I-1) \leq 2\epsilon.$$

Therefore, the probability on the left hand side of (20) is bounded below by

$$P(|X_i - x_{i0}| \leq \epsilon^2, X_i \geq \tau, i = 1, \cdots, I - 1)$$

$$\geq \quad \frac{\Gamma(\alpha I)}{(\Gamma(\alpha))^I} \prod_{i=1}^{I-1} \int_{\max((x_{i0}-\epsilon^2),\tau)}^{\min((x_{i0}+\epsilon^2),1)} x_i^{\alpha-1} dx_i.$$

Since $\alpha < 1$, we can lower bound the integrand by 1 and the interval of integration contains at least an interval of length $\epsilon^2 - \tau$. Therefore, the result above can be further lower bounded by

$$\frac{\Gamma(\alpha I)}{(\Gamma(\alpha))^I} (\epsilon^2 - \tau)^{I-1} \geq \frac{\Gamma(\alpha I)}{(\Gamma(\alpha))^I} (\epsilon^2 - \tau)^I.$$

This finishes the proof. $\qquad \square$

Now, we are ready to derive lower bounds for the prior probability mass on $\Theta_I$'s when $I$ varies within a certain range. Before stating the result, we want to briefly review the assumptions we made in Section 3. First, in terms of approximation error, we assume that for any $f_0 \in \mathcal{F}_0$, there exists a sequence of $f_I \in \Theta_I$, such that $A_1 I^{-r} \leq \min_{g \in \Theta_I} \rho(g, f) \leq \rho(f_I, f) \leq A_2 I^{-r}$ for some positive constants $A_1$ and $A_2$ (If the lower bound does not hold, we can always obtain a faster concentration rate). Second, we imposed a moment condition on $\mathcal{F}_0$. For any $f \in \mathcal{F}_0$, we assume that $\int f^2 < \infty$. At last, given a partition of size $I$, the weights on the subregions within the partition follow a Dirichlet distribution truncated from below, with the truncation parameter $\tau = D I^{-\eta}$ $(D, \eta > 0)$. Under these three assumptions, we will derive the lower bound in the lemma below.

**Lemma B.4.** *Assume that $f_0 \in \mathcal{F}_0$. $\Pi$ is the prior probability specified in Section 2. Let $t_{n,I} = \epsilon_{n,I}^2 = \frac{I \log I}{n/\log n}$. Take $I = n^{\frac{1}{2r+1}}$, we have*

$$\mathbb{P}_{f_0}^n \Big( \int_{\Theta_I} \prod_{j=1}^n \frac{f(Y_i)}{f_0(Y_i)} d\Pi(f)$$

$$\leq \frac{1}{2} \Pi(\Theta_I) \exp(-2nt_{n,I} - c^* I \log I - 4\omega I \log n - I \log \Gamma(\alpha)) \Big)$$

$$\leq \frac{2}{nt_{n,I}},$$

*where $\omega = \max(1, 1/2r)$, and $c^*$ is the constant introduced in Section 2.2.*

*Proof.* Let $S_{n,I} = \{f \in \Theta_I : K(f_0, f) \leq t_{n,I}, V(f_0, f) \leq t_{n,I}\}$. From lemma B.1, we have the bound

$$\mathbb{P}_{f_0}^n \Big( \int_{\Theta_I} \prod_{j=1}^n \frac{f(Y_i)}{f_0(Y_i)} d\Pi(f) \leq \frac{1}{2} \Pi(S_{n,I}) e^{-2nt_{n,I}} \Big) \leq \frac{2}{nt_{n,I}}. \tag{21}$$

Next step, we will search a lower bound for $\Pi(S_{n,I})$. The way to approach this is to find a subset of $S_{n,I}$ to which we can apply Lemma B.3. Our argument is as the following.

Define $\tilde{S}_{n,I} = \{f \in \Theta_I : K(f_0, f) \leq t_{n,I}, \mathbb{E}_{f_0}\big( (\log \frac{f_0(Y)}{f(Y)})^2 \big) \leq t_{n,I}, f \geq \tau \}$, where $\tau$ is a truncation parameter which will be specified later. Note that $\mathbb{E}_{f_0}\big( (\log \frac{f_0(Y)}{f(Y)})^2 \big) \geq V(f_0, f)$, we have $\tilde{S}_{n,I} \subset S_{n,I}$. From (19), we know that

$$W_{n,I} := \{f \in \Theta_I : \rho(f_0, f) \leq \frac{L\epsilon_{n,I}}{\log \frac{M_\delta}{\epsilon_{n,I}}}, f \geq \tau \} \subset \tilde{S}_{n,I}.$$

Set $\tau$ to be $DI^{-\eta}$ with $\eta > \max\{2, 4r\}$, then $M_\delta = O(I^{\delta\eta} \int f_0^{(1+\delta)})$. Furthermore,

$$\frac{\epsilon_{n,I}}{\log \frac{M_\delta}{\epsilon_{n,I}}} = O\left( \frac{\left( \frac{I \log I}{n/\log n} \right)^{1/2}}{\log \left( I^{\delta\eta} \int f_0^{(1+\delta)} (\frac{n/\log n}{I \log I})^{1/2} \right)} \right)$$

$$= O\left( \left( \frac{I \log I}{n \log n} \right)^{1/2} \right). \tag{22}$$

Under the assumptions that $I = n^{\frac{1}{1+2r}}$, there exists $f_I \in \Theta_I$, such that $\rho(f_0, f_I) < \frac{L\epsilon_{n,I}}{\log \frac{M_\delta}{\epsilon_{n,I}}}$. If we define

$$\tilde{W}_{n,I} := \{f \in \Theta_I : \rho(f, f_I) \leq \frac{L\epsilon_{n,I}}{\log \frac{M_\delta}{\epsilon_{n,I}}} - \rho(f_0, f_I), f \geq \tau \},$$

by triangle inequality, we know that $\tilde{W}_{n,I} \subset W_{n,I}$. Together with the previous result, we claim that there exists a constant $L'$, such that

$$\tilde{B}_{n,I} := \{f \in \Theta_I : \rho(f, f_I) \leq L' \left( \frac{I \log I}{n \log n} \right)^{1/2}, f \geq \tau \} \subset \tilde{W}_{n,I},$$

Next, from the fact $\rho^2(f, g) \leq \|f - g\|_{L_1}$, we have

$$B_{n,I} := \{f \in \Theta_I : \|f_I - f\|_{L_1} \leq \frac{L'^2 I \log I}{n \log n}, f \geq \tau \} \subset \tilde{B}_{n,I}.$$

Note that $\Pi(B_{n,I}) = \Pi(\Theta_I)\Pi(B_{n,I}|\Theta_I)$. Assume that $f_I$ is supported by the binary partition $\{\Omega_{i0}\}_{i=1}^I$. Let $F_0 = \{f \in \Theta_I : f = \sum_{i=1}^I \frac{\theta_i}{|\Omega_{i0}|} \mathbb{1}_{\Omega_{i0}}, \theta_i \geq 0, \sum_{i=1}^I \theta_i = 1\}$ be the collection of all the density functions in $\Theta_I$ which are supported by the same binary partition as $f_I$. Then

$$\Pi(B_{n,I}|\Theta_I) \geq \Pi(B_{n,I}|F_0)\Pi(F_0|\Theta_I) \geq \exp(-c^* I \log I)\Pi(B_{n,I}|F_0). \tag{23}$$

Now we apply Lemma B.3 to bound $\Pi(B_{n,I}|F_0)$ from below. We will works with an $L_1$-ball with radius $(\frac{L'^2 I \log I}{n \log n})^\omega$, where $\omega$ is chosen to be $\max(1, 1/2r)$. We can always assume that $L' < 1$, otherwise we can work with a ball shringking to zero at a faster rate instead. Obviously, this ball is contained in $B_{n,I}$. When $I = n^{\frac{1}{2r+1}}$, we have $(\frac{L'^2 I \log I}{n \log n})^\omega < \frac{1}{I}$. Under the assumptions $\eta > \max(2, 4r)$, we know that when $I/n^{\gamma_1} = o(1)$ with $\gamma_1 = \frac{2\omega}{2\omega+\eta}$, $DI^{-\eta} = o((\frac{I \log I}{n \log n})^{2\omega})$. By setting $x_{i0}$ in the lemma to be the probability mass on $\Omega_{i0}$ under $f_I$, we have

$$
\begin{aligned}
\Pi(B_{n,I}|F_0) &\geq \frac{\Gamma(\alpha I)}{(\Gamma(\alpha))^I}((\frac{L'^2 I \log I}{2n \log n})^{2\omega} - DI^{-\eta})^I \\
&\geq \exp(-I \log \Gamma(\alpha) - 4\omega I \log n).
\end{aligned}
\tag{24}
$$

Combine (21), (23) and (24) together, we get the desired result. $\qquad\square$

## C  Proof of Theorem 3.1

In this section, we will combine the upper bound in Section A and the lower bound in Section B together to derive the posterior concentration rate.

*Proof of Theorem 3.1.* Let $\epsilon_n = n^{-\frac{r}{2r+1}}(\log n)^{2+\frac{1}{2r}}$ and $\eta_{n,I} = \left(\frac{I(\log I)^{1/r+1}}{n/\log n}\right)^{1/2}$. First, we divide the items in (9) into three blocks. We define

$$
I_{Num} = \sum_{I=1}^{N_1-1} \int_{\{f:\rho(f,f_0)\geq M\epsilon_n\}\cap\Theta_I} \prod_{j=1}^{n} \frac{f(Y_j)}{f_0(Y_j)} d\Pi(f),
$$

$$
II_{Num} = \sum_{I=N_1}^{N_2} \int_{\{f:\rho(f,f_0)\geq M\epsilon_n\}\cap\Theta_I} \prod_{j=1}^{n} \frac{f(Y_j)}{f_0(Y_j)} d\Pi(f),
$$

$$
III_{Num} = \sum_{I=N_2+1}^{n/\log n} \int_{\{f:\rho(f,f_0)\geq M\epsilon_n\}\cap\Theta_I} \prod_{j=1}^{n} \frac{f(Y_j)}{f_0(Y_j)} d\Pi(f),
$$

where $N_1 = D_1 n^{\frac{1}{2r+1}}(\log n)^{-\frac{1}{r}}$ and $N_2 = D_2 n^{\frac{1}{2r+1}}(\log n)^2$.

We deal with each block in the numerator separately. Roughly speaking, when $I$ is small, the approximation error to $f_0$ dominates, and these items can be bounded by the Hellinger distance between $f$ and $f_0$. The items in the middle range can be bounded by controlling the metric entropy of $\Theta_I$. The items in the last block are negligible because the prior probability decays to zero fast.

We assume that there exists a sequence of $f_I \in \Theta_I$, such that $A_1 I^{-r} \leq \min_{g\in\Theta_I} \rho(g, f) \leq \rho(f_I, f) \leq A_2 I^{-r}$ for some positive constants $A_1$ and $A_2$. Let $N_3 = D_3 n^{\frac{1}{2r+1}}(\log n)^{-\frac{2}{r}-\frac{1}{2r^2}}$. With an appropriately chosen $D_3$, when $I < N_3$, $A_1 I^{-r}$ is greater than $M\epsilon_n$. Therefore,

$$
\begin{aligned}
&\sum_{I=1}^{N_3-1} \int_{\{f:\rho(f,f_0)\geq M\epsilon_n\}\cap\Theta_I} \prod_{j=1}^{n} \frac{f(Y_j)}{f_0(Y_j)} d\Pi(f) \\
&= \sum_{I=1}^{N_3-1} \int_{\Theta_I} \prod_{j=1}^{n} \frac{f(Y_j)}{f_0(Y_j)} d\Pi(f).
\end{aligned}
\tag{25}
$$

When $N_3 \leq I < N_1$, given that $A_1 I^{-r} < M\epsilon_n$, we have

$$
\begin{aligned}
&\sum_{I=N_3}^{N_1-1} \int_{\{f:\rho(f,f_0)\geq M\epsilon_n\}\cap\Theta_I} \prod_{j=1}^{n} \frac{f(Y_j)}{f_0(Y_j)} d\Pi(f) \\
&\leq \sum_{I=N_3}^{N_1-1} \int_{\{f:\rho(f,f_0)\geq A_1 I^{-r}\}\cap\Theta_I} \prod_{j=1}^{n} \frac{f(Y_j)}{f_0(Y_j)} d\Pi(f).
\end{aligned}
\tag{26}
$$

Combine (25) and (26) together and apply Lemma A.4 by setting $\delta_{n,I}$ to be $A_1 I^{-r}$, we obtain

$$
\begin{aligned}
I_{Num} &\leq \sum_{I=1}^{N_1-1} \int_{\{f:\rho(f,f_0)\geq A_1 I^{-r}\}\cap\Theta_I} \prod_{j=1}^{n} \frac{f(Y_j)}{f_0(Y_j)} d\Pi(f) \\
&\leq \sum_{I=1}^{N_1-1} \Pi(\Theta_I) \exp(-A_1 n I^{-2r}) \\
&\leq \left(\sum_{I=1}^{N_1-1} \exp(-2A_1 n I^{-2r})\right)^{1/2}.
\end{aligned}
$$

The last line is based on the Caucy-Schwarz inequality. Now, we will estimate the order of the summation in the last line. In order to simplify the notation, we will discuss the order of $\sum_{I=1}^{N_1-1} \exp(-\frac{2A_1 n}{I^{2r}})$ in detail.

We know that the mass is centered around $I = N_1 - 1$. Power series expansion around that point gives

$$
\sum_{I=1}^{(1-\epsilon)N_1} \leq (1-\epsilon)N_1 \exp\left(-\frac{2A_1 n}{((1-\epsilon)N_1)^{2r}}\right),
$$

which is a lower order term compared to the last term in the summation and thus does not contribute significantly to the summation. Let $1 - \delta = \frac{I}{N_1}$, expand

$$
(1-\delta)^{-2r} = 1 + 2r\delta + \binom{-2r}{2}\delta^2 + o(\delta^2).
$$

$$
\begin{aligned}
\sum_{I=(1-\epsilon)N_1}^{N_1-1} \exp(-\frac{2A_1 n}{I^{2r}}) &\leq \int_{(1-\epsilon)N_1}^{N_1} \exp(-\frac{2A_1 n}{x^{2r}})dx \\
&\sim \int_0^\epsilon \exp\left(-2\frac{A_1}{D_1^{2r}} n^{\frac{1}{2r+1}}(\log n)^2 (1-\delta)^{-2r}\right) N_1 d\delta \\
&\sim \int_0^\epsilon \exp\left(-2\frac{A_1}{D_1^{2r}} n^{\frac{1}{2r+1}}(\log n)^2 (1+2r\delta+o(\delta))\right) N_1 d\delta \\
&\sim \frac{1}{(\log n)^{1/r+2}} \exp\left(-2\frac{A_1}{D_1^{2r}} n^{\frac{1}{2r+1}}(\log n)^2\right).
\end{aligned}
$$

Therefore

$$
I_{Num} \leq (\log n)^{-1-\frac{1}{2r}} \exp(-\frac{A_1}{D_1^{2r}} n^{\frac{1}{2r+1}}(\log n)^2). \tag{27}
$$

From Lemma A.4, we know that if the result applies for $\delta_{n,I}$, then it also applies to $M\eta_{n,I} > \delta_{n,I}$. We have that when $N_1 \leq I \leq N_2$,

$$
\begin{aligned}
II_{Num} &\leq \sum_{I=N_1}^{N_2} \int_{\{f:\rho(f,f_0)\geq M\eta_{n,I}\}\cap\Theta_I} \prod_{j=1}^{n} \frac{f(Y_j)}{f_0(Y_j)} d\Pi(f) \\
&\leq \sum_{I=N_1}^{N_2} \exp(-\lambda I \log I) \exp(-M^2 I(\log I)^{1+\frac{1}{r}} \log n) \\
&\leq \left(\sum_{I=N_1}^{N_2} \exp(-2\lambda I \log I)\right)^{1/2} \left(\sum_{I=N_1}^{N_2} \exp\left(-2M^2 I(\log I)^{1+\frac{1}{r}} \log n\right)\right)^{1/2} \\
&\sim \exp\left(-M^2 n^{\frac{1}{2r+1}}(\log n)^2\right),
\end{aligned}
$$

where the last line is obtained by integration by part.

For the last block $III_{Num}$, we have

$$III_{Num} \quad \leq \quad \sum_{I=N_2+1}^{n/\log n} \int_{\Theta_I} \prod_{j=1}^n \frac{f(Y_j)}{f_0(Y_j)} d\Pi(f)$$

$$\sim \quad \exp\left(-n\int f_0 \log(f_0)\right) \sum_{I=N_2+1}^{n/\log n} \int_{\Theta_I} \prod_{j=1}^n f(Y_j) d\Pi(f). \qquad (28)$$

If we use $x_I$ to represent a partition of size $I$, and $\mathcal{X}_I$ to denote the collection of all binary partitions of size $I$, then the integral in (28) can be divided into the integral over each partition as the following:

$$III_{Num}$$

$$\lesssim \quad \exp\left(-n\int f_0 \log(f_0)\right) \sum_{I=N_2+1}^{n/\log n} \sum_{x_I \in \mathcal{X}_I} \int_{\theta_1,\dots,\theta_I} \prod_{j=1}^n f(Y_j|\theta_1,\dots,\theta_I,x_I)$$

$$\times \Pi(\theta_1,\dots,\theta_I|x_I)\Pi(x_I)d\theta_1\dots d\theta_I$$

$$\lesssim \quad \exp\left(-n\int f_0 \log(f_0)\right)$$

$$\times \sum_{I=N_2+1}^{n/\log n} \frac{\exp(-\lambda I \log I)}{T_I} \sum_{x_I \in \mathcal{X}_I} \frac{D(\alpha+n_1,\dots,\alpha+n_I)}{D(\alpha,\dots,\alpha)} \prod_{i=1}^I \frac{1}{|\Omega_i|^{n_i}},$$

The last inequality is obtained by integrating out $\theta_i$'s. Now, we focus on the part inside the summation, and apply Stirling's approximation to the gamma function,

$$\frac{D(\alpha+n_1,\dots,\alpha+n_I)}{D(\alpha,\dots,\alpha)} \prod_{i=1}^I \frac{1}{|\Omega_i|^{n_i}}$$

$$= \quad \exp\left( \log\Gamma(\alpha I) - I\log\Gamma(\alpha) + \sum_{i=1}^I \log\Gamma(\alpha+n_i) \right.$$

$$\left. - \log\Gamma(\alpha I + n) + \sum_{i=1}^I n_i \log\frac{1}{|\Omega_i|} \right)$$

$$\lesssim \quad \exp\left( (\alpha I - 1)\log(\alpha I - 1) - (\alpha I - 1) - I\log\Gamma(\alpha) - (\alpha I + n - 1)\log(\alpha I + n - 1) + \alpha I + n - 1 \right.$$

$$+ \sum_{i:n_i\geq 1} \left( (\alpha+n_i-1)\log(\alpha+n_i-1) - (\alpha+n_i-1) + \frac{1}{2}\log(\alpha+n_i-1) + \log\sqrt{2\pi} \right)$$

$$\left. + \sum_{i:n_i\geq 1} n_i \log\frac{1}{|\Omega_i|} \right), \qquad (29)$$

Let $C(\alpha) = \log\sqrt{2\pi} + 1 - \log\Gamma(\alpha) - \alpha$, then

$$(29)$$

$$\lesssim \quad \exp\left( (\alpha I - 1)\log\frac{\alpha I - 1}{\alpha I + n - 1} - n\log(\alpha I + n - 1) + C(\alpha)I + (\alpha - \frac{1}{2})I\log\frac{n+(\alpha-1)I}{I} \right.$$

$$\left. + \sum_{i:n_i\geq 1} n_i \log\frac{n_i}{|\Omega_i|} \right). \qquad (30)$$

Next, we will find an upper bound of $\sum_{i:n_i \geq 1} n_i \log \frac{n_i}{|\Omega_i|}$. Given a partition $\{\Omega_i\}_{i=1}^{I}$, define $\mu_i = \int_{\Omega_i} f_0$, $\hat{\mu}_i = n_i/n$, $\nu_i = \mu_i/|\Omega_i|$, and $f_{x_I} = \sum_{i=1}^{I} \nu_i \mathbb{1}_{\Omega_i}$. Then,

(30)

$$= \exp\left(-n \log \frac{\alpha I + n - 1}{n} + C(\alpha)I + \sum_{i:n_i \geq 1} n_i \log \frac{\hat{\mu}_i}{\mu_i} + \sum_{i:n_i \geq 1} n_i \log \nu_i\right)$$

$$\leq \exp\left(-n \log \frac{\alpha I + n - 1}{n} + C(\alpha)I + n \int f_0 \log(f_0) - nK(f_0, \hat{f}_{x_I})\right.$$

$$\left. +n \sum_{i:n_i \geq 1} \hat{\mu}_i \log \frac{\hat{\mu}_i}{\mu_i} + n \sum_{i:n_i \geq 1} (\hat{\mu}_i - \mu_i) \log(\nu_i)\right)$$

Next, we will use large deviations to study the last two items in the summation. Applying a result in [3] (Corollary 2.5 in the paper), we have

$$\mathbb{P}_{f_0}^{n}\left(\sum_{i:n_i \geq 1} \hat{\mu}_i \log \frac{\hat{\mu}_i}{\mu_i} \geq \frac{\eta I \log I}{n}\right) \lesssim \exp(-\eta I \log I),$$

where $\eta$ is a constant. We can choose $\eta$ such that this result holds for all partitions of size $I$ with probability tending to 1.

Based on this, we know that when $I$ is between $N_2$ and $n/\log n$, the integral over each partition is bounded given that $\lambda$ is large enough. Indeed, the condition on $\lambda$ is that $\lambda > \eta$ and $\eta > c^*$. For example, we can set $\lambda = 3$ and $\eta = 2$. If we plug in this result into the summation, we have

$$III_{Num} \lesssim \sum_{I=N_2}^{n/\log n} \exp(-I \log I)$$

$$\leq \exp(-D_2 n^{\frac{1}{2r+1}}(\log n)^2).$$

Therefore

(9)

$$\lesssim \frac{(\log n)^{-1-\frac{1}{2r}} \exp(-(A_1/D_1^{2r})n^{\frac{1}{2r+1}}(\log n)^2) + \exp(-M^2 n^{\frac{1}{2r+1}}(\log n)^2 + \exp(-D_2 n^{\frac{1}{2r+1}}(\log n)^2))}{\sum_{I=1}^{\infty} \int_{\Theta_I} \prod_{j=1}^{n} \frac{f(Y_j)}{f_0(Y_j)} d\Pi(f)}$$

$$\leq \frac{(\log n)^{-1-\frac{1}{2r}} \exp(-(A_1/D_1^{2r})n^{\frac{1}{2r+1}}(\log n)^2) + \exp(-M^2 n^{\frac{1}{2r+1}}(\log n)^2) + \exp(-D_2 n^{\frac{1}{2r+1}}(\log n)^2)}{\frac{1}{2}\exp\left(-\frac{2}{2r+1}n^{\frac{1}{2r+1}}(\log n)^2 - (\frac{c^*}{2r+1} + 4\omega)n^{\frac{1}{2r+1}}\log n - n^{\frac{1}{2r+1}}(\log \Gamma(\alpha) + 1)\right)},$$

where the last inequality is obtained by applying Lemma B.4 to the space $\Theta_I$ with $I = n^{\frac{1}{2r+1}}$. The last line goes to zero when $A_1/D_1$, $M^2$ and $D_2$ are all greater than $\frac{2}{2r+1}$.

Therefore, we have

$$\Pi\left(f : \rho(f, f_0) \geq M\epsilon_n | Y_1, \cdots, Y_n\right) \leq \exp\left(-bn^{\frac{1}{2r+1}}(\log n)^2\right),$$

with probability tending to 1, where $b$ is a positive constant. This concludes the proof. $\qquad\square$

# D  Proof of Corollaries

## D.1  High-dimensional Haar basis

We first provide more details about high-dimensional Haar basis. In one dimension, the Haar wavelet's mother wavelet function is

$$\psi(y) = \begin{cases} 1 & \text{if } 0 \leq y < 1/2, \\ -1 & \text{if } 1/2 \leq y < 1, \\ 0 & \text{otherwise.} \end{cases}$$

And its scaling function is

$$\phi(y) = \begin{cases} 1 & \text{if } 0 \le y < 1, \\ 0 & \text{otherwise.} \end{cases}$$

Here, we take the two-dimensional case to illustrate how the system is built. This construction can be extended to high dimensional cases as well.

The two-dimensional scaling function is defined to be

$$\phi\phi(y^1, y^2) := \phi(y^1)\phi(y^2),$$

and three wavelet functions are

$$\phi\psi(y^1, y^2) := \phi(y^1)\psi(y^2),$$
$$\psi\phi(y^1, y^2) := \psi(y^1)\phi(y^2),$$
$$\psi\psi(y^1, y^2) := \psi(y^1)\psi(y^2).$$

If we use a superscript $l$ to index the scaling level of the wavelet function and subscripts $i$ and $j$ ($i$ and $j$ can be $0, 1, \cdots, 2^l - 1$) to denote the horizontal and vertical translations respectively, then the scales and translates of the three wavelet functions $\phi\psi$, $\psi\phi$ and $\psi\psi$ are defined to be

$$\phi\psi_{ij}^l(y^1, y^2) := (\sqrt{2})^{2 \cdot l} \phi\psi(2^l y^1 - i, 2^l y^2 - j),$$
$$\psi\phi_{ij}^l(y^1, y^2) := (\sqrt{2})^{2 \cdot l} \psi\phi(2^l y^1 - i, 2^l y^2 - j),$$
$$\psi\psi_{ij}^l(y^1, y^2) := (\sqrt{2})^{2 \cdot l} \psi\psi(2^l y^1 - i, 2^l y^2 - j).$$

These functions together with the single scaling function $\phi\phi$ define the two-dimensional Haar wavelet basis $\mathbf{\Psi}$.

## D.2 Spatial sparsity

**Lemma D.1.** *Suppose $f_0$ is a $p$-dimensional density function. $g_0 = \sqrt{f_0}$ satisfies the condition (10). Then there exists a sequence of $f_I \in \Theta_I$, such that $\rho(f_0, f_I) \lesssim I^{-(q-1/2)}$, or equivalently, $\rho(f_0, f_I) \le cI^{-(q-1/2)}$, where $c$ may depend on $q$ and $p$ but not $I$.*

*Proof.* See [5] proof of Lemma 4.1. $\qquad\square$

The proof of Corollary 3.2 follows directly from this lemma and Theorem 3.1.

## D.3 Density functions of bounded variation

Let $\Lambda$ be the set of indices for the wavelet basis. Each element in $\Lambda$ is a pair of scale and location parameters. We will denote by $\Sigma_N$ the spaces consisting of $N$-term approximation in the Haar system, in other words,

$$\Sigma_N := \{\sum_{\lambda \in E} c_\lambda \psi_\lambda : E \subset \Lambda, |E| \le N\},$$

where $|E|$ denotes the cardinality of the discrete set $E$.

First, we cite a result from [1]. It provides a bound for the approximation rate to a function of bounded variation by $\Sigma_N$.

**Lemma D.2.** *If $f \in BV(\Omega)$ has mean value zero on $\Omega$, we have*

$$\inf_{g \in \Sigma_N} \|f - g\|_{L_2(\Omega)} \le CN^{-1/2}V_\Omega(f), \tag{31}$$

*with $C = 2592(3\sqrt{5} + \sqrt{3})$.*

Assume $f_0$ is a density function on $\Omega$ of bounded variation. By subtracting the mean, we can always assume that $\sqrt{f_0}$ has mean value zero over $\Omega$. For the square root of $f_0$, applying the lemma above, we can find an $N$-term approximation $g$ in the Haar system, such that $\|\sqrt{f_0} - g\|_{L_2(\Omega)} \lesssim N^{-1/2}$. Translating this inequality into the size of partition, we reach the conclusion that for a density function in $BV(\Omega)$, we can find an approximation in $\Theta_I$, such that $\rho(f_0, f_I) \lesssim I^{-1/2}$. Corollary 3.3 follows.

### D.4 Hölder space

**Lemma D.3.** *If $\sqrt{f_0}$ is Hölder continuous (when $p = 1$) or mixed-Hölder continuous (when $p \geq 2$) with regularity parameter $\beta \in (0, 1]$, then there exists a sequence of $f_I \in \Theta_I$, such that $\rho(f_0, f_I) \lesssim I^{-\beta/p}(\log I)^{p/2}$.*

*Proof.* See [5] proof of Lemma 6.1. $\qquad\square$

The proof of Corollary 3.4 follows from this.