[Reviews · NeurIPS 2017]

Reviewer 1



Note: Below, I use [#M] for references in the main paper and [#S] for references in the supplement, since these are indexed differently. Summary: This paper proposes and analyzes a Bayesian approach to nonparametric density estimation. The proposed method is based on approximation by piecewise-constant functions over a binary partitioning of the unit cube, using a prior that decays with the size of the partition. The posterior distribution of the density is shown to concentrate around the true density f_0, at a rate depending on the smoothness r of f_0, a measure in terms of how well f_0 can be approximated by piecewise-constant functions over binary partitionings. Interestingly, the method automatically adapts to unknown r, and r can be related to more standard measures of smoothness, such as Holder continuity, bounded variation, and decay rate of Haar basis coefficients. As corollaries, posterior concentration rates are shown for each of these cases; in the Holder continuous case, this rate is minimax optimal. Major Comments: The theoretical results of this paper appear quite strong to me. In particular, the results on adaptivity (to unknown effective dimension and smoothness) are quite striking, especially given that this seems an intrinsic part of the estimator design, rather than an additional step (as in, e.g., Lepski's method). Unfortunately, I'm not too familiar with Bayesian or partition-based approaches to nonparametric density estimation, and my main issue is that the relation of this work to previous work isn't very clear. Looking through the supplement, it appears that most of proofs are based on the results of [9M] and [14M], so there is clearly some established closely related literature. I'd likely raise my score if this relation could be clarified, especially regarding the following two questions: 1) Is the prior introduced in Section 2.2 novel? If so, how does it differ from similar prior work (if such exists), and, if not, which aspects, precisely, of the paper are novel? 2) Much of the proof of Theorem 3.1 appears to depend on results from prior work (e.g., Lemmas A.2, A.4, B.1, and B.2, and Theorem A.3 are from [4S] and [6S]). Corrolaries 3.2, 3.3, and 3.4 all follow by combining Theorem 3.1 with results from [4S] and [1S] that relate the respective smoothness condition to smoothness in terms of approximability by piecewise-constant functions on binary partitions. Thus, much of this work appears to be a re-working of previous work, while Lemma B.4 and the remainder of Theorem 3.1 appear to be the main contributions. My question is: at a high level, what were the main limitations of the previous results/proof techniques that had to be overcome to prove Theorem 3.1, and, if it's possible to summarize, what were the main innovations required to overcome these? Minor Comments: Lines 36-44: It would help to mention here that rates are measured in Hellinger divergence. Line 40-41: (the minimax rate for one-dimensional Hölder continuous function is (n/log n)^{−\beta/(2\beta+1)}): If I understand correctly, the log factors stem from the fact that \beta is unknown (I usually think of the minimax rate as n^{−\beta/(2\beta+1)}, when \beta is treated as known). If this is correct, it would help to mention this here. Line 41: small typo: "continuous function" should be "continuous functions" Lines 60-72: There's an issue with the current notation: As written, \Omega_2,...,\Omega_I aren't well-defined partition by this recursive procedure. If we split \Omega_j at step i, then \Omega_j should be removed from the list of partitions and replaced by two smaller partitions. I think I understand what is meant (Figure 1 is quite clear), but I don't see a great way to explain this with concise mathematical notation - the options I see are (a) describing the process as a tree with a set at each node, and then taking all the leaves of the tree or (b) using a pseudocode notation where the definition of \Omega_j can change over the course of the recursive procedure. Line 185: I believe the minimax rate for the bounded variation class is of order n^(-1/3) (see, e.g., Birgé, Lucien. "Estimating a density under order restrictions: Nonasymptotic minimax risk." The Annals of Statistics (1987): 995-1012.) Perhaps this is worth mentioning? Lines 157-175: Section 3.1 considers the case of a weak-\ell^p constraint on the Haar basis coefficients of the density. The paper calls this a spatial sparsity constraint. I feel this is misleading, since the sparsity assumption is over the Haar basis coefficients, rather than over spatial coordinates (as in [1M]). As a simple example, the uniform distribution is extremely sparse in the Haar basis, but is in no way spatially concentrated. I believe this is actually a smoothness assumption, since Haar basis coefficients can be thought of as identifying the discontinuities of a piecewise constant approximation function. Indeed, the weak-\ell^p constraint on the Haar coefficients is roughly equivalent to a bound on the Besov norm of the density (see Theorem 5.1 of Donoho, David L. "De-noising by soft-thresholding." IEEE transactions on information theory 41.3 (1995): 613-627.) Line 246: small typo: "based adaptive partitioning" should be "based on adaptive partitioning". Line 253: Citation [1M] is missing the paper title. I believe the intended citation is "Abramovich, F., Benjamini, Y., Donoho, D. L., & Johnstone, I. M. (2006). Special invited lecture: adapting to unknown sparsity by controlling the false discovery rate. The Annals of Statistics, 584-653."

Reviewer 2



Summary: The paper presents a Bayesian approach to density estimation. The approach first constructs a binary partition tree on the domain and then uses a piecewise constant estimator which is constant on each cell of the partition. The main theorem gives the rate of posterior concentration in terms of a parameter r which specifies how well a class is approximated by the partitioning scheme. The authors bound r for different smoothness classes for the density. While I did not read the proofs in the appendix, I found the technical exposition substantial and interesting. On the truncation of the Dirichlet prior: - Can you elaborate what exactly the technical condition in Line 127 is? Ideally, you should include this detail in the main text. - I am assuming that the posterior uses the truncated prior? How difficult is it to compute this? - Lines 131-132 and 148-149 suggest picking a large eta, but what do you lose if you pick a very large eta? Does this, for instance, affect the rate? (e.g. a 1/eta term in front of the rate?) - If the Dirichlet coefficients are bounded away from 0, then it means that you are putting non-zero mass across the entire domain. In instances where the support of f0 is a very small set, it appears that such an estimate might be very bad? Can you explain why you get the same rates even for such an f0? Some comments/questions/suggestions: - It might make sense to separate the discussion about [13] at the end of section 2.2 into a separate subsection or paragraph. When reading it at first, I had assumed that it was an extension of the prior. - Can you generalise the definition of bounded variation to higher dimensions? - The Holder definition for Multivariate densities used in Section 3.3 is somewhat nonstandard, see for e.g Defintion 3 in [1]. Can you bound r for this definition too? - I thought that the observations in lines 196-199 was quite interesting. Experiments: - The main theorems talk about the posterior, but not any specific estimate? What is the estimate used in the experiments? Is it the mean of the posterior? References [1] Kandasamy et al, Nonparametric Von Mises Estimators for Entropies Divergences and Mmutual Informations.

Reviewer 3



The paper presents the concentration rate of partition based Bayesian density estimation. The paper proposes a suitable prior for densities on a binary adaptive partition, and shows explicitly the concentration rates in some specific situations, including the density being spatially sparse, belonging to the space of bounded variation, and being Holder continuous. The theory is well explained and a few relevant concepts are well reviewed. The theoretical contribution is remarkable and numerical examples are provided to illustrate and support the methodological contributions.